# Effects of implementing non-nutritive sucking on oral feeding progression and outcomes in preterm infants: A systematic review and meta-analysis

Shuliang Zhao[1,2‡], Huimin Jiang[2‡], Yiqun Miao[3], Wenwen Liu[4], Yanan Li[2], Yuanyuan Zhang[2], Aihua Wang[2]*, Xinghui Cui[1]*

1 Nursing Department, Affiliated Hospital of Shandong Second Medical University, Weifang, China, 2 School of Nursing, Shandong Second Medical University, Weifang, China, 3 School of Nursing, Capital Medical University, Beijing, China, 4 Xiangya School of Nursing, Central South University, Changsha, China

‡ SZ and HJ are contributed equally to this work and co-first authors.
* Wangaihua64@163.com (AW); lcyxycxh@163.com (XC)

## Abstract

**Data Availability Statement:** All relevant data are within the manuscript and its Supporting Information files.

### Background

Preterm infants have imperfect neurological development, uncoordinated sucking-swallowing-breathing, which makes it difficult to realize effective oral feeding after birth. How to help preterm infants achieve complete oral feeding as soon as possible has become an important issue in the management of preterm infants. Non-nutritive sucking (NNS), as a useful oral stimulation, can improve the effect of oral feeding in preterm infants. This review aimed to explore the effect of NNS on oral feeding progression through a meta-analysis.

### Methods

We systematically searched PubMed, CINHAL, Web of Science, Embase, Cochrane databases, China's National Knowledge Infrastructure (CNKI), Wanfang and VIP database from inception to January 20, 2024. Search terms included 'non-nutritive sucking' 'oral feeding' and 'premature.' Eligibility criteria involved randomized controlled studies in English or Chinese. Studies were excluded if they were reviews, case reports, or observational studies from which valid data could not be extracted or outcome indicators were poorly defined. The meta-analysis will utilize Review Manager 5.3 software, employing either random-effects or fixed-effects models based on observed heterogeneity. We calculated the mean difference (MD) and 95% confidence interval (CI) for continuous data, and estimated pooled odds ratios (ORs) for dichotomous data. Sensitivity and publication bias analyses were conducted to ensure robust and reliable findings. We evaluated the methodological quality of randomized controlled trials (RCTs) utilizing the assessment tool provided by the Cochrane Collaboration.

**Funding:** The author(s) received no specific funding for this work.

**Competing interests:** The authors have declared that no competing interests exist.

## Results

A total of 23 randomized controlled trials with 1461 preterm infants were included. The results of the meta-analysis showed that NNS significantly shortened time taken to achieve exclusive oral feeding (MD = -5.37,95%CI = -7.48 to-3.26, $p$<0.001), length of hospital stay (MD = -4.92, 95% CI = -6.76 to -3.09, $p$<0.001), time to start oral feeding(MD = -1.41, 95% CI = -2.36 to -0.45, $p$ = 0.004), time to return to birth weight(MD = -1.72, 95% CI = -2.54 to -0.91, $p$<0.001). Compared to the NNS group, the control group had significant weight gain in preterm infants, including weight of discharge (MD = -61.10, 95% CI = -94.97 to -27.23, $p$ = 0.0004), weight at full oral feeding (MD = -86.21, 95% CI = -134.37 to -38.05, $p$ = 0.0005). In addition, NNS reduced the incidence of feeding intolerance (OR = 0.22, 95% CI = 0.14 to 0.35, $p$<0.001) in preterm infants.

## Conclusion

NNS improves oral feeding outcomes in preterm infants and reduces the time to reach full oral feeding and hospitalization length. However, this study was limited by the relatively small sample size of included studies and did not account for potential confounding factors. There was some heterogeneity and bias between studies. More studies are needed in the future to validate the effects on weight gain and growth in preterm infants. Nevertheless, our meta-analysis provides valuable insights, updating existing evidence on NNS for improving oral feeding in preterm infants and promoting evidence-based feeding practices in this population.

## Introduction

Preterm birth, defined as delivery before 37 weeks of gestation, poses a significant global health burden and is recognized as a primary risk factor for children mortality under the age of five [1, 2]. Preterm birth is linked to a spectrum of adverse short- and long-term outcomes, including compromised health and growth during infancy, intellectual and psychiatric disabilities later in life, and an increased risk of chronic diseases developing at an earlier age [3]. The implications of preterm birth extend beyond the initial period following birth and can have lasting impacts on the individual's overall well-being across their lifespan. Oral feeding and swallowing problems in preterm infants are common medical problems in neonatal intensive care units (NICUs) and families of preterm infant [4]. Preterm birth presents many challenges to infants, including neurodevelopmental immaturity, physiological instability, and disturbed behavioral states [5, 6]. Preterm infants are prone to oral feeding difficulties due to delayed development of oral motor skills and poor suck-swallow-breathe coordination [5–7]. In addition, sucking and swallowing dysfunction in preterm infants affects the availability of nutrients and may affect their growth, development and neurological function [8, 9]. The transition from tube-feeding to full oral feeding is a huge challenge for infant caregivers [10]. Early difficulties with oral feeding can affect the ability of preterm infants to achieve independent oral feeding, prolong hospitalization, and cause long-term feeding problems such as malnutrition, growth retardation, and cognitive deficits [5, 11, 12]. Therefore, providing optimal nutritional feeding support for preterm infants, improving sucking and feeding behaviors in infants less

than 30 weeks of age, and increasing the ability to feed independently through the mouth are key to the early care and survival of preterm infants [13].

Non-nutritive sucking (NNS) and nutritive sucking (NS) serve as indices of an infant's oral motor skills and feeding behavior [14]. The Neonatal Oral-Motor Assessment Scale (NOMAS) is an instrument utilized for appraising the oral motor abilities of infants [15]. By using the NOMAS scale, healthcare professionals can identify potential oral motor problems in infants and provide appropriate intervention and support. NNS refers to a pacifier is placed in an infant's mouth to increase sucking action, but there is no milk or other fluid intake [16]. NNS creates an oral feeding experience and promotes self-organization and soothing. Oral feeding not only necessitates the components of NNS but also requires suck–swallow–breathe coordination to manage a bolus for consumption [17]. Numerous studies have demonstrated that NNS offers clinical advantages, such as enhancing feeding performance and reducing the duration of hospital stays, and accelerating the transition to oral feeding [18, 19]. Effective NNS interventions not only decrease the incidence of adverse events like oxygen desaturation, apnoea, and bradycardia but may also lessen long-term issues such as feeding or eating aversion [20].

The meta-analysis by Schwartz et al. [21] in 1987 indicated that non-nutritive sucking shortened the time to first bottle feed and reduced hospital stay duration. Both the systematic reviews by Pinelli et al. [22] in 2005 and Barlow et al. [10] in 2008 concurred that NNS could enhance feeding skills in preterm infants and reduce hospitalization time. Foster et al.'s [23] meta-analysis in 2016 demonstrated significant effects of NNS on the transition from tube to full oral feeding, from the start of oral feeding to full oral intake, and on hospital stay duration. Despite a consensus reached by previously published meta-analyses on the promotion of feeding performance in preterm infants through NNS, numerous randomized controlled trials have been conducted in recent years. Therefore, we decided to carry out a more comprehensive and systematic literature review and meta-analysis. The objective of our meta-analysis was to examine the effects of implementing NNS versus no NNS on the initiation of oral feeding, the attainment of full oral feeding, hospital stay duration, and weight progression in preterm infants. This study aimed to synthesize existing evidence to provide an updated understanding of the efficacy of NNS interventions in neonatal intensive care units (NICUs).

## Materials and methods

### Systematic search and strategy

The review protocol was developed using the Cochrane Handbook for Systematic Reviews of Interventions. Our meta-analysis was conducted according to the preferred reporting items in the guide of systematic review and meta-analysis (PRISMA) [24]. The PRISMA checklist is presented in S1 Checklist. A comprehensive search was performed in MEDLINE (via PubMed), CINHAL, Web of science, Embase, Cochrane Library, CNKI, VIP and Wanfang databases to identify randomized controlled trials (RCT) from inception to January 20, 2024. The following search terms were used:("Sucking Behavior" OR "Pacifiers" OR (non-nutritive AND suck*) OR (non-nutritive sucking) OR (Non-nutritional sucking) OR pacifier OR dummy OR soother OR nipple) AND ("Infant, Newborn" OR "Infant, Premature" OR newborn OR neonate OR neonatal OR premature OR (very low birth weight) VLBW OR (low birth weight) LBW OR (extremely low birth weight) ELBW) AND (randomized controlled trial OR controlled clinical trial OR RCT). Detailed search strategies for each database are shown in S1 Table. Only research involving human beings is included in the selection. Moreover, language limitations are English or Chinese. Our meta-analysis was registered with the PROSPERO database (CRD42023457646).

## Inclusion and exclusion criteria

Studies included in this systematic review and meta-analysis must meet the following inclusion criteria: ① Preterm infants with gestational age < 37 weeks and birth weight < 2500g; ② Randomized controlled studies; ③ The language is published in Chinese or English; ④Preterm infants in the intervention group received NNS (The intervention can occur before, during or after gavage feeding. NNS involving the use of a pacifier or nipple.) and preterm infants in the control group were implemented on routine feeding. ⑤At least one of the parameters was included in the outcome measures: Time taken to achieve exclusive oral feeding, Length of hospital stay, Time to start oral feeding, Weight of discharge, Weight at full oral feeding, Time to return to birth weight, Gastrointestinal complications. Time taken to achieve exclusive oral feeding was defined as when the infant ingests all nutrient volumes in a 24hour period without any gavage [25]. Length of hospitalization is described as the number of days from admission to hospital discharge [26]. Time to start oral feeding is defined as transition from gavage feeding to oral feeding, measured in days [27]. Weight of discharge is described as infant's weight on discharge from hospital, measured in grams [28]. Weight at full oral feeding is described as infant's weight at full oral intake, measured in grams [29]. Time to return to birth weight is described as the time it takes for a newborn to lose weight after birth and then gradually return to its birth weight, measured in days [30]. Feeding intolerance (FI) is a disorder of milk digestion after enteral feeding, resulting in abdominal distension, vomiting and gastric retention [27], and the diagnostic criteria are based on the Clinical Guidelines on Feeding Intolerance in Preterm Infants [31], with the incidence rate of FI serving as the unit of measurement.

Any study that met the exclusion criteria was excluded from the systematic review and meta-analysis: ① Reviews, case reports, observational studies or animal trials. ② Full text was unavailable ③ Studies with insufficient information for extraction data ④ Unclear or inappropriate definition of exposure/results ⑤ NNS was implemented in the control group.

## Data extraction

Two researchers (ZSL and JHM) screened through the literature independently and completed the extraction of data. If there is a disagreement, a third researcher (MYQ) should be called in to adjudicate. Initial screening was done first by reading the title and abstract, followed by selecting the final compliant literature by reading the full text. The extracted information includes the following aspects: first author's name, year of publication, country, sample size, details of the intervention, and the data of outcome indicators. If the units of the outcome indicators in the study are inconsistent, they should be converted to harmonized units prior to subsequent processing.

## Methodological quality assessment

We assessed the methodological quality of RCTs using the Cochrane risk of bias tool [32], which included seven evaluation items: sequence generation, allocation concealment, blinding of participants, outcome evaluator, incomplete outcome data, and selective outcome reporting. The studies were evaluated for unclear, low or high risk of bias. In addition, we entered the assessment data into Cochrane Review Manager software (Revman version 5.3, Copenhagen: The Nordic Cochrane Centre, The Cochrane Collaboration, Denmark) to obtain a risk of bias assessment table. This was done independently by two assessors (LWW and LYN), and when disagreements arose, they were resolved in consultation with a third researcher (ZSL)until agreement was reached.

## Data synthesis and analysis

The data was analyzed by Review Manager 5.3 software. The mean difference (MD) and confidence interval (CI) of 95% were calculated for continuous data, while the pooled odds ratios (ORs) and confidence interval (CI) of 95% were estimated for dichotomous data. Heterogeneity was assessed by using $I^2$ statistics. Random-effects models were employed in instances where there was substantial heterogeneity among the studies ($I^2 > 50\%$), whereas fixed-effects models were utilized when the heterogeneity was low ($I^2 \leq 50\%$) [33]. Sensitivity analysis was conducted by changing the analytical model, i.e., selecting the opposite model according to the original analytical model and conducting the combined analysis of effect sizes again. If the results of the two models were consistent, it indicated that the sensitivity was lower, suggesting that the results of the meta-analysis were stable, and vice versa, suggesting that the sensitivity was higher and the results of the meta-analysis were unstable. After individual studies were sequentially excluded, the data were reanalyzed and sensitivity analyses were performed by comparing the excluded results with the original results. Subgroup analyses were used when necessary to address heterogeneity. Funnel plot was performed to evaluate publication bias. $p < 0.05$ was considered to be statistically significant.

# Results

## Search results

The initial search yielded 3238 potentially eligible records for eight databases: PubMed (n = 367), EMBASE (n = 339), CINAHL (n = 169), Web of Science (n = 177), Cochrane Library (n = 671), CNKI (n = 349), Wanfang database (n = 708) VIP (n = 458). Following the removal of 1385 duplicate records, a total of 1,853 unique records were assessed. Based on title and abstract screening, 72 studies were selected for full-text review. At this stage, 44 studies were excluded due to various reasons: being reviews, case reports, conference abstracts, animal trials (n = 30), absence of full text (n = 3), or having unclear or inadequate outcome definitions (n = 11). Twenty—eight studies [19, 26–30, 34–55] were identified for inclusion in the systematic review (qualitative synthesis). Four studies [51–54] did not meet our inclusion criteria because the control preterm infants underwent breast or pacifier NNS; data could not be extracted from one additional study [55], so only 23 studies were included in the meta-analysis. The PRISMA flow diagram in Fig 1 shows the study selection process.

Table 1 shows the detailed characteristics of the included literature. These articles included were published from 1982 to 2023. Ten studies [26, 27, 34–41] were conducted in China, three studies [42–44] in the United States, three studies [45–47] in Iran, five studies [28, 30, 48–50] in Turkey, and one each in Canada [29] and Brazil [19]. Figs 2 and 3 depict the risk-of-bias assessment in minute detail. Only six studies [19, 26, 45–47, 50] showed a low risk of bias, and seventeen studies [27–30, 34–44, 48, 49] showed a higher risk of bias. Because it is difficult to implement double-blinding for NNS implementation, only six [19, 26, 45–47, 50] of the included papers were better blinded.

## Meta analysis results

### Primary outcomes

**Time taken to achieve exclusive oral feeding.** Time taken to achieve exclusive oral feeding(days) was included as an outcome between the two groups in fifteen studies [19, 26–30, 34, 35, 39, 42, 44–48] that involved 942 premature infants. We observed that the time to transition to full oral feeding in the NNS group was significantly shorter compared to the control group

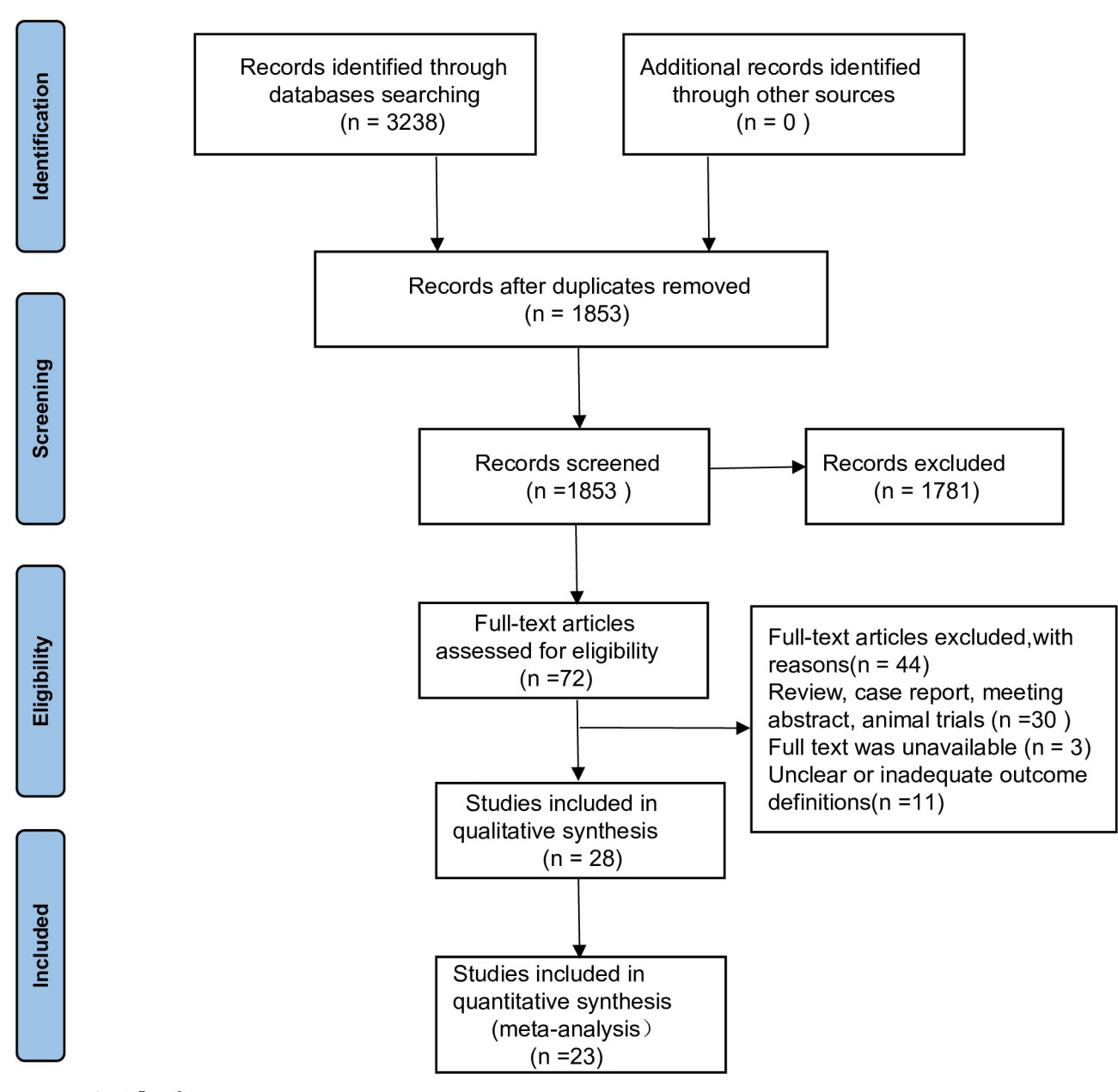

**Fig 1. PRISMA flow diagram.**

(MD = -5.37,95%CI = -7.48 to -3.26, $p<0.001$) (Fig 4). Heterogeneity between studies was high ($I^2$ = 93%, $p < 0.001$), so a random-effect model was used.

**Length of hospital stay.** Seventeen studies [19, 27–30, 34, 36, 37, 39, 42–49] involving 1006 premature infants examined the effect of NNS on length of hospital stay and contributed data to the meta-analysis. Meta-analysis showed a statistically significant shorter length of hospital stay for infants in the NNS compared to the control infants (MD = - 4.92, 95% CI = -6.76 to -3.09, $p<0.001$). Significant heterogeneity existed among these studies ($I^2$ = 83%, $p<0.001$). Subgroup analyses was performed based on gestational age of preterm infants (Fig 5). Heterogeneity was significantly reduced in preterm infants with gestational age >33 weeks ($I^2$ = 0%,

**Table 1. Characteristics of the included studies.**

| Study | Year | Country | Number | | Intervention | | Outcomes |
|---|---|---|---|---|---|---|---|
| | | | NNS | Control | NNS group | Control group | |
| Shaki [45] | 2022 | Iran | 50 | 50 | Sucking on a pacifier 3 times/day followed by gavage for ten days of intervention. | routine ward care | ①②③ |
| Ostadi [46] | 2021 | Iran. | 15 | 15 | provided with NNS twice a day, for 10 days during two consecutive weeks | not provided with any intervention | ①②③ |
| Say [30] | 2018 | Turkey | 45 | 45 | Use the pacifiers for 15 minutes before and after feeding, 4 times/day. | not provided with any intervention | ①②④⑥⑦ |
| Fucile [29] | 2018 | Canada | 16 | 15 | The intervention was a 15-minute program consisting of 5 minutes of preparatory oral stimulation of the cheeks and lips, 5 minutes of tongue exercises, and 5 minutes of NNS. 1x/day for 10 days. | 15 minutes of pseudo-intervention | ①②⑤ |
| Kaya [49] | 2017 | Turkey | 34 | 36 | Sucking 3 times a day to soothe | Not using pacifiers | ②③④⑤ |
| Asadollahpour [47] | 2015 | Iran | 11 | 11 | NNS 3 times/day for 15 minutes each time. Place little finger in infant's mouth and stroke palate for 5 min/times to elicit sucking. | pseudo-intervention | ①②④ |
| Zhang [34] | 2014 | China | 25 | 27 | Suck on the pacifier 7–8 times/day for 5 minutes. | not provided with any intervention | ①②③⑤ |
| Yildiz [48] | 2012 | Turkey | 30 | 30 | 3x/day sucking on a pacifier until oral feeding is initiated | Routine gavage feeding without NNS | ①②④ |
| Rocha [19] | 2007 | Brazil | 49 | 49 | NNS 15min/dose for 10 days until oral feeding is initiated | 15 minutes of pseudo-intervention | ①②③④⑤ |
| Lau [42] | 2012 | USA | 25 | 23 | NNS, 15 minutes/day, 5 days per week. | Routine feeding care | ①②③④ |
| Field [43] | 1982 | USA | 30 | 27 | infants in the treatment group were given a pacifier during all tube feedings. | Control infants were tube-fed only until oral feeding | ②③ |
| Bernbaum [44] | 1983 | USA | 15 | 15 | Infants in the intervention group were given rubber nipples for NNS during the entire feeding process | Control infants were tube-fed only until oral feeding | ①② |
| Xu [27] | 2022 | China | 57 | 57 | Breast milk olfactory stimulation combined with NNS. | Routine feeding care | ①②③④⑦ |
| Feng [35] | 2022 | China | 44 | 46 | Early swallowing function training combined with NNS | Routine feeding care | ①③⑦ |
| Ling [36] | 2008 | China | 26 | 21 | NNS was given before each nasogastric tube feeding and sucked for 10min every 2-3h/time. | No NNS | ②③⑥⑦ |
| Zhang [37] | 2003 | China | 22 | 21 | NNS every 3h with 10min sucking for 2 weeks. | No NNS | ②③⑥ |
| Yue [38] | 2003 | China | 18 | 20 | Preterm infants were given a non-porous rubber nipple to suck for 5 min before, during and after each nasogastric tube feeding for 2 weeks. | No NNS | ③⑥⑦ |
| Ma [39] | 2019 | China | 43 | 43 | NNS was given for 5 min before each feeding until the preterm infant was able to complete 8 sucks/d for 3 h/suck for 7 d of continuous intervention. | No NNS | ①② |
| Jiang [40] | 2009 | China | 50 | 48 | Preterm infants were given a non-porous rubber nipple to suck for 5 min before, during and after each feeding, 7–8 times/24h, for a period of 1 week. | Routine gavage feeding without NNS | ③⑥⑦ |
| Kang [41] | 2009 | China | 36 | 36 | Sucking on a non-porous rubber nipple for 10 min before each feeding, 8 times/d. | Routine gavage feeding without NNS | ③⑥⑦ |
| Lyu [26] | 2014 | China | 32 | 31 | Implemented an oral stimulation program consisting of 12 minutes of oral stimulation and 3 minutes of NNS. administered 15–30 minutes before feeding, once a day. | Routine feeding care in the NICU | ①④⑤⑥ |
| Berber Çiftci [50] | 2024 | Turkey | 47 | 47 | NNS was continued until discharge for 5 days a week, once a day, until transition from tube feeding to full oral feeding. | Routine feeding care | ④⑤ |
| Calik [28] | 2019 | Turkey | 14 | 14 | Using a pacifier, each feeding lasts about 10 minutes and continues until complete oral feeding. | not given the pacifiers | ①②④ |
| Li [51] | 2023 | China | 74 | 74 | Non-nutritive breast sucking once a day, 5 min each time, and sucking pacifier for 5 min before each tube feeding. | Suck pacifier for 5 min before each tube feeding | ① |
| John [52] | 2019 | India | 4 | 5 | standard care and suckle on the mother's emptied breast for 5 to 10 minutes 3 times a day | Use fingers for NNS | ②④ |

*(Continued)*

**Table 1.** (Continued)

| Study | Year | Country | Number | | Intervention | | Outcomes |
|---|---|---|---|---|---|---|---|
| | | | NNS | Control | NNS group | Control group | |
| Khodagholi [53] | 2018 | Iran | 16 | 16 | NNS paired with olfactory stimuli was administered during the initial 5 minutes of each gavage feeding, thrice daily for 10 days | NNS was administered as in the intervention group with untreated cotton pads | ③④ |
| Fucile [54] | 2021 | Canada | 16 | 17 | Empty Breast Group NNS, once a day for 15 min | For the NNS on a pacifier Group, once a day for 15 min | ①②③④⑤ |
| Kamhawy [55] | 2014 | Egypt | 23 | 24 | Placed pacifiers in preterm infants' mouths for 15 minutes of NNS around feeding times, NNS for four consecutive feedings per day for the total of 10 continuous days. | Not using pacifiers | ③④ |

Note: NNS: Non-nutritive sucking

①Time taken to achieve exclusive oral feeding(days) ②Length of hospital stay(days) ③Time to start oral feeding(days) ④Body weight of discharge(g) ⑤Weight at full oral feeding (g) ⑥Time to return to birth weight(days) ⑦ Feeding intolerance

$p = 0.94$), and statistical differences between the NNS and control groups remained after grouping.

**Time to start oral feeding.** Fourteen studies [19, 27, 34–38, 40–43, 45, 46, 49] involving 957 premature infants included time to start oral feeding as an outcome. Compared to the control group, we found a statistically significant reduction in transition from gavage to start oral feeding in the NNS group (MD = -1.41, 95% CI = -2.36 to -0.45, $p = 0.004$) (Fig 6), and there was greater heterogeneity ($I^2 = 59\%$, $p = 0.003$). Heterogeneity was reduced after one article [42] was removed from the sensitivity analysis ($I^2 = 40\%$, $p = 0.07$).

## The secondary outcomes

**Weight of discharge.** Ten studies [19, 26–28, 30, 42, 47–50] including 687 patients were included to assess weight of discharge. The result of random-effect model demonstrated that there was not a significant difference in weight at discharge between the two groups (MD = -22.61, 95% CI = -84.92 to -39.69, $p = 0.48$). Heterogeneity was significantly reduced after excluding two articles [28, 47] from the sensitivity analysis ($I^2 = 0\%$, $p = 0.45$). This adjustment revealed a statistically significant difference between the two groups (MD = -61.10, 95% CI = -94.97 to -27.23, $p = 0.004$) (Fig 7).

**Time to return to birth weight.** Time to return to birth weight was included as an outcome between the two groups in seven studies [26, 30, 36–38, 40, 41] that involved 451 infants. Compared to the control group, preterm infants in the NNS group had a shorter time to return to birth weight (MD = -1.72, 95% CI = -2.54 to -0.91, $p < 0.001$) (Fig 8). Significant heterogeneity existed among these studies ($I^2 = 51\%$, $p = 0.06$), so the random-effect model was used.

**Weight at full oral feeding.** Six studies [19, 26, 29, 34, 49, 50] involving 408 premature infants reported the weight at full oral feeding as an outcome. The meta-analysis revealed a statistically significant difference between the two groups, with preterm infants in the NNS group exhibiting reduced weight as compared to control group (MD = -86.21, 95% CI = -134.37 to -38.05, $p = 0.0005$) (Fig 9). There was no significant heterogeneity between these studies ($I^2 = 0\%$, $p = 0.87$), so a fixed-effects model was used.

**Feeding intolerance.** Seven studies [27, 30, 35, 36, 38, 40, 41] involving 549 premature infants reported the feeding intolerance as an outcome. According to our meta-analysis, feeding intolerance in preterm infants were significantly less in the NNS group than in the control group(OR = 0.22, 95% CI = 0.14 to 0.35, $p < 0.001$) (Fig 10). There was no significant heterogeneity between these studies ($I^2 = 0\%$, $p = 0.86$), so a fixed-effects model was used.

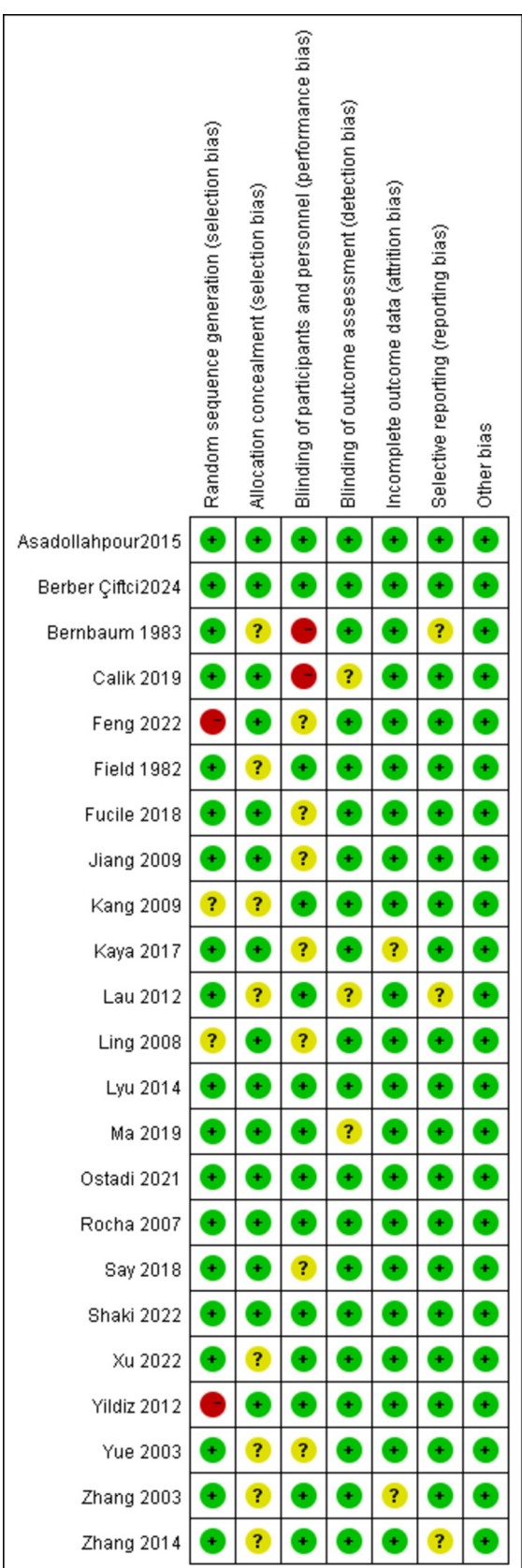

**Fig 2. Risk of bias graph about each risk of bias item presented as percentages across all included studies.**

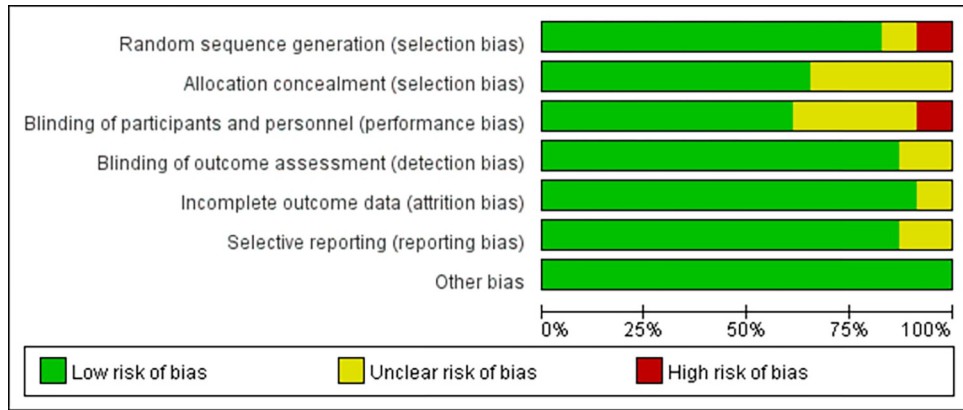

**Fig 3. Summary of risk of bias for each trial.**

## Sensitivity analysis and publication bias

Sensitivity analysis by converting the effect sizes of the fixed-effects model and the random-effects model to each other revealed that the differences in the combined effect sizes of the two were small, indicating that the results of the meta-analysis were generally stable (Table 2). In addition, the source of heterogeneity was further explored using a study-by-study approach to exclude individual studies. We drew funnel plots for the indicators of the number of 10 or more articles included in the literature and evaluated the publication bias. The results showed that the funnel plots were all largely symmetrical, with no obvious publication bias (S1 Fig).

# Discussion

Recent investigations have demonstrated that NNS not only enables nasogastric tube-fed preterm infants to transition to oral feeding as soon as possible, but also has a significant effect on both behavioral patterns and growth and development of preterm infants [53, 56]. Antecedent study has identified the advantages of NNS in improving sucking skills, shortening transition to full breastfeeding, and time to hospital discharge in preterm infants [49]. It has also been shown that the use of pacifiers can negatively affect the health of infants and lead to nipple confusion [57]. At the same time, NNS provided by a pacifier might elicit different physiological, pharyngeal, and esophageal motility events, potentially affecting gastroesophageal reflux [58].

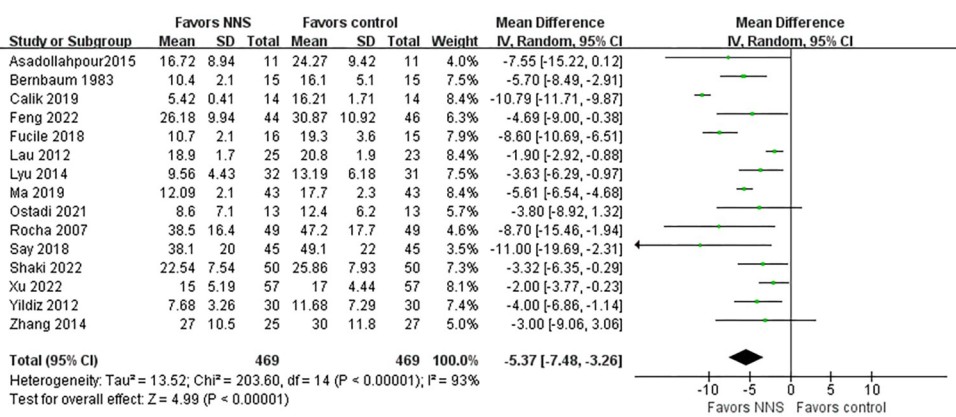

**Fig 4. Forest plot for time taken to achieve exclusive oral feeding.**

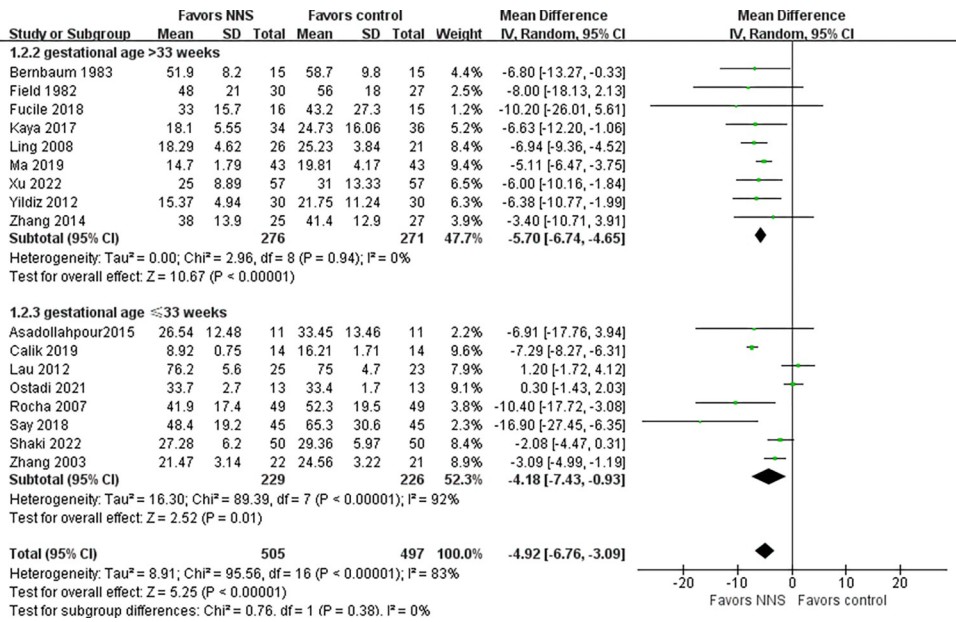

**Fig 5. Forest plot for length of hospital stay.**

Therefore, the present study conducted a more comprehensive and systematic meta-analysis aimed at investigating the impact of NNS on the effectiveness of oral feeding in preterm infants. Infants in the NNS group demonstrated a significantly reduced duration necessary to achieve exclusive oral feeding, a shorter length of hospital stay, an earlier commencement of oral feeding, and a quicker recovery to birth weight when compared to the control group. Preterm infants in the NNS group weighted less at hospital discharge and at full oral feeding. In addition, this review demonstrated that NNS reduced the incidence of feeding intolerance in preterm infants.

First of all, our meta-analysis found that the premature infants in NNS group had significantly shorter time taken to reach exclusive oral feeding and time to start oral feeding than those in the control group, which was consistent with the Cochrane analysis of Foster et al. [23]. Preterm infant perioral and oral receptors are highly sensitive to stimulation, mechanical stimulation of the lips or tongue can cause a non-specific reflex in the orbicularis oris muscle,

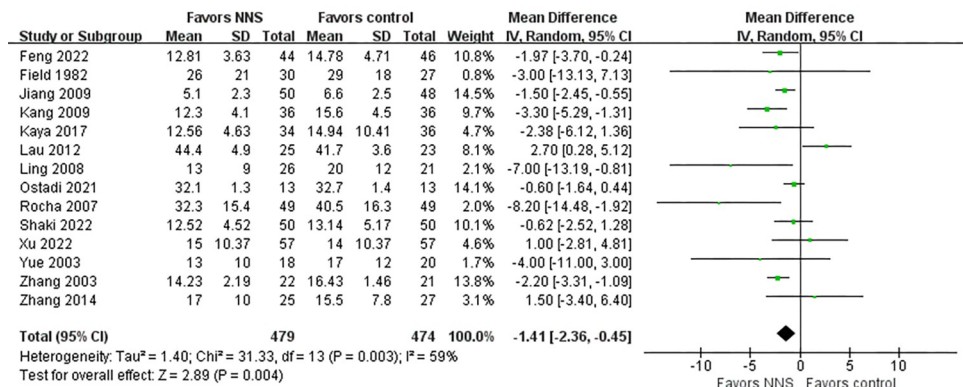

**Fig 6.** Forest plot for time to start oral feeding.

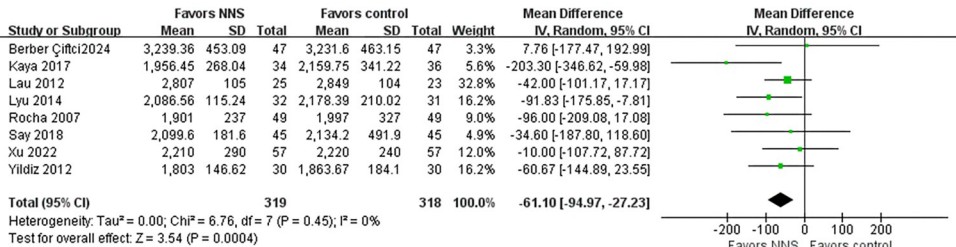

**Fig 7. Forest plot for weight of discharge.**

thus initiating sucking behavior [59]. The sensory experience generated by NNS can trigger sucking in preterm infants through the input-feedback mechanism of the central pattern generator (CPG), prompting an increase in the number of active sucks, a faster rate of sucking, an increase in the negative pressure of sucking, and a shortening of the stopping time, thus promoting the maturation of the sucking reflex [59]. In addition, NNS motor training of preterm infants' oral muscles promotes the coordination of sucking, swallowing, and respiration, thus accelerating the transition from tube-feeding to transoral feeding [56]. The meta-analysis by Tolppola et al. [60] in 2022 was also in agreement with our findings, confirming the reduction in time from gavage to total mouth feeding in preterm infants in the NNS group, and suggesting that the use of NNS in hospitalized preterm infants is beneficial and should be carried out in clinical practice.

Adequate nutrition increases the rate of development of premature infant bodies, leading to shorter hospital stays. Consistent with the results of a previous meta-analysis [10, 21, 22], our study demonstrates that NNS significantly reduces the length of hospitalization for preterm infants. Say et al. [30], Kaya et al. [49] and Shaki et al. [45] studies also showed that preterm infants in the NNS group were discharged earlier, and these results are consistent with the present study. Analyzing the reasons, this may be related to the fact that NNS not only promotes the maturation of sucking reflex in preterm infants and shortens the time from tube feeding to complete oral feeding [61], but also prompts hepatic, biliary and pancreatic activities through the vagus nerve, regulates gastrointestinal peptide levels, and stimulates gastrointestinal growth and development and maturation, which in turn improves feeding tolerance in preterm infants and shortens the length of hospital stay [37]. Clinical experience has demonstrated that the length of hospitalization of preterm infants is largely dependent on the time taken to achieve complete oral feeding. However, the reality is that when a preterm infant is discharged from the NICU after a stay in the NICU is related to a number of complex factors, such as: stable weight gain, mature and stable cardiorespiratory function as well as assessment of nutritional risk, neurodevelopmental and neurobehavioral assessments, and more

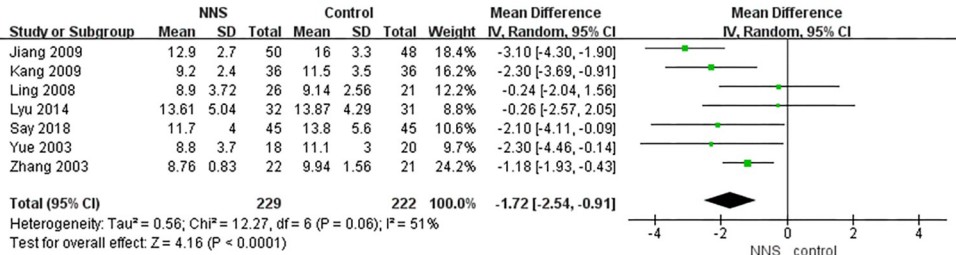

**Fig 8. Forest plot for time to return to birth weight.**

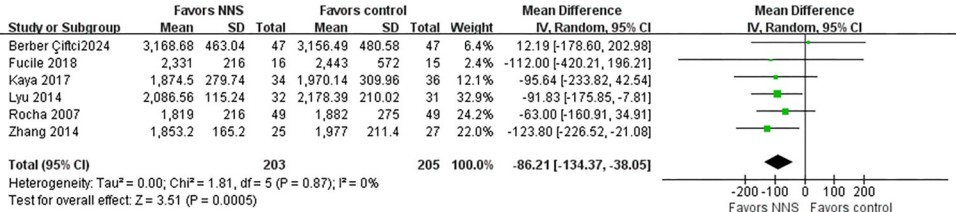

**Fig 9. Forest plot for weight at full oral feeding.**

[62]. It is also important to ensure that the family and community health care system is adequately prepared to care for the infant safely outside of the hospital after discharge.

Our study showed that preterm infants in the NNS group had lower body weights at hospital discharge compared to the control group, a finding that is consistent with the findings of Lyu et al. [26] and different from those of Asadollahpour et al. [47] and Calik et al. [28]. The difference in discharge weight between the different studies in preterm infants may be related to the baseline weight of the study. The improvement in weight of preterm infants at discharge by NNS may also be related to the following factors: NNS can stimulate the vagus nerve excitation in the oral cavity of preterm infants, and then stimulate the secretion of gastric motility and gastrin, regulate gastrointestinal motility, promote the development of gastrointestinal function, so that preterm infants eat more [63], but also cause vagus nerve excitation, accelerate the digestion and absorption of nutrients, and promote the growth of physical development [61]. In addition, NNS training can stimulate the orofacial and lingual muscles of preterm infants, improve the strength of oropharyngeal muscles, improve the function of swallowing reflexes, enhance the sucking ability of children, shorten the time of feeding through the mouth, and promote weight gain [64].Other studies [27, 30, 49] have also found that preterm infants in the NNS and control groups had similar weight measurements at discharge, and the difference was not statistically significant, but it does not indicate that NNS does not promote body mass growth in preterm infants. This is because there are many other factors involved in the weight gain and growth of preterm infants. The variability in the effect of NNS on discharge weight of preterm infants between studies may be related to the different methods of intervention, and there are differences in the timing as well as duration of NNS use that can affect the effectiveness of the intervention, so this result should be interpreted with caution.

Successful oral feeding has been defined as the ability of preterm infants to complete the prescribed amount of milk and gain the expected weight. However, our study found that the mean weight of preterm infants in the NNS group was lower than that of the control group at the time of achieving independent oral feeding, which is consistent with the findings of Lyu

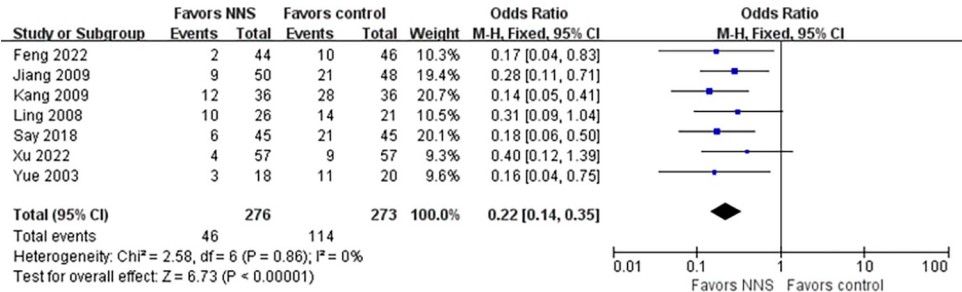

**Fig 10. Forest plot for feeding intolerance.**

**Table 2. Results of sensitivity analysis.**

| Outcome indicator | Fixed effect model *OR(95%CI)* | random effect model *OR(95%CI)* |
|---|---|---|
| Time taken to achieve exclusive oral feeding | -5.91 (-4.73, -3.64) | -5.37 (-7.48, -3.26) |
| Length of hospital stay | -4.90 (-5.49, -4.32) | -4.92 (-6.76, -3.09) |
| Time to start oral feeding | -1.39 (-1.88, -0.90) | -1.41 (-2.36, -0.45) |
| Weight of discharge | -61.10 (-94.97, -27.23) | -61.10 (-94.97, -27.23) |
| Weight at full oral feeding | -86.21 (-134.37, -38.05) | -86.21(-134.37, -38.05) |
| Time to return to birth weight | -1.67 (-2.17, -1.17) | -1.72 (-2.54, -0.91) |
| Feeding intolerance | 0.22 (0.14, 0.35) | 0.22 (0.14, 0.35) |

et al. [26]. In contrast, the findings of Rocha et al. [19] found that NNS had no significant effect on the weight of preterm infants at the time of independent mouth-feeding. The study by Rocha et al. [19] documented the average weekly weight gain, whereas most of the studies that we included measured the rate of weight gain over the entire period of hospitalization. Different measurements can affect the results of the study, in addition to the fact that the weight of preterm infants is influenced by other factors. The maximum degree of physiologic weight loss and the time to regain birth weight reflect nutrient availability and physical development. In our study, we found that the difference in time to regain birth weight between the two groups of preterm infants was statistically significant, which is consistent with the results of Zhang et al. [37] and Yue et al. [38]. NNS accelerates the maturation of the sucking reflex in preterm infants, improves the ability to suck, and facilitates oral gratification, which leads to an increase in the amount of milk ingested by the infants. In addition, the pituitary gland increased the secretion of various hormones, especially growth hormone secretion reached the highest level, which is conducive to the growth and development of preterm infants [65]. Hwang et al. [66] concluded that NNS can modulate pre-feeding behavior in preterm infants, increase feeding efficiency during the first 5 min of feeding, and help infants regulate pre-feeding agitation.

In the present study, the incidence of feeding intolerance in preterm infants in the NNS group was significantly less than those in the control group, which is consistent with the findings of Say et al. [30] and Feng et al. [35]. The possible explanation for this is that NNS helps to establish rhythmic sucking and swallowing patterns in preterm infants, promotes gastrointestinal growth and development and the maturity of gastrointestinal function, and reduces feeding intolerance and gastrointestinal complications by stimulating the sensory nerve fibers in the oral cavity and stimulating the G-cells to release gastric motility, gastrin, and gastric acid secretion [67].

This review conducted an extensive literature search to refine the evidence base supporting the use of NNS to improve oral feeding outcomes in preterm infants and to promote evidence-based feeding practices. The findings of this study underscore the benefits of NNS for oral feeding in preterm infants, which can inform the development of structured oral feeding programs in NICUs. Such programs have the potential to provide preterm infants with positive early feeding experiences, thereby enhancing their overall care and development. Moving forward, there is a need for large-scale multicenter randomized controlled trials to further investigate the efficacy of NNS in improving oral feeding in preterm infants, and to assist in the development of individualized feeding patterns and evidence-based clinical guidelines for oral feeding in preterm infants.

Nonetheless, this study has several limitations that must be acknowledged. First, although we conducted a relatively comprehensive search, we may have missed some eligible studies. Only articles written in English or Chinese were included, and gray literature was not considered. Second, some articles did not clearly describe randomization and blindness, which may

lead to biased reports. Finally, there was some heterogeneity in the interventions, with some using a combination of oral stimulation and NNS, and some using only NNS, but as the results were similar across all studies, this should not have been an issue in the analysis.

## Conclusions

In conclusion, NNS significantly reduces the time required for preterm infants to achieve exclusive oral feeding, as well as shortening their length of hospital stay and the time to both initiate oral feeding and return to birth weight. Although we observed that the weight at discharge and at the achievement of full oral feeding was lower in the NNS group compared to the control group, this does not negate the potential for NNS to positively influence weight gain in preterm infants. In the future, it is necessary to further explore the effect of NNS on weight gain of preterm infants, and to establish a standardized evidence-based management method for oral feeding of preterm infants. Importantly, NNS also appears to decrease the incidence of feeding intolerance, which supports its consideration for widespread use in NICUs.

## Supporting information

**S1 Checklist. PRISMA 2020 checklist.**
(DOCX)

**S1 Table. Search strategy.**
(DOCX)

**S1 Fig. Funnel plot of publication bias.** ((a): Time taken to achieve exclusive oral feeding; (b): Length of hospital stay; (c): Time to start oral feeding).
(ZIP)

## Author Contributions

**Conceptualization:** Shuliang Zhao, Huimin Jiang, Yiqun Miao.

**Data curation:** Shuliang Zhao, Huimin Jiang, Wenwen Liu, Yanan Li.

**Formal analysis:** Shuliang Zhao, Huimin Jiang, Yiqun Miao.

**Funding acquisition:** Yuanyuan Zhang, Aihua Wang, Xinghui Cui.

**Investigation:** Shuliang Zhao, Huimin Jiang, Wenwen Liu, Yanan Li.

**Methodology:** Shuliang Zhao, Huimin Jiang, Yiqun Miao, Wenwen Liu, Yanan Li.

**Project administration:** Yuanyuan Zhang, Aihua Wang, Xinghui Cui.

**Resources:** Aihua Wang, Xinghui Cui.

**Software:** Shuliang Zhao, Huimin Jiang.

**Validation:** Shuliang Zhao, Yiqun Miao, Aihua Wang.

**Writing – original draft:** Shuliang Zhao, Huimin Jiang.

**Writing – review & editing:** Shuliang Zhao, Aihua Wang, Xinghui Cui.

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
