## [Decision Letter · Decision Letter 0]

16 Jan 2024

PONE-D-23-30243Effectiveness of non-nutritive sucking on oral feeding of preterm infants: a systematic review and meta-analysisPLOS ONE

Dear Dr. cui,

Thank you for submitting your manuscript to PLOS ONE. After careful consideration, we feel that it has merit but does not fully meet PLOS ONE’s publication criteria as it currently stands. Therefore, we invite you to submit a revised version of the manuscript that addresses the points raised during the review process. Please submit your revised manuscript by Mar 01 2024 11:59PM. If you will need more time than this to complete your revisions, please reply to this message or contact the journal office at plosone@plos.org. Please include the following items when submitting your revised manuscript:A rebuttal letter that responds to each point raised by the academic editor and reviewer(s). You should upload this letter as a separate file labeled 'Response to Reviewers'.A marked-up copy of your manuscript that highlights changes made to the original version. You should upload this as a separate file labeled 'Revised Manuscript with Track Changes'.An unmarked version of your revised paper without tracked changes. You should upload this as a separate file labeled 'Manuscript'.

We look forward to receiving your revised manuscript.

Kind regards,

Mona Nabulsi, MD, MS

Academic Editor

PLOS ONE

Journal Requirements:

Reviewers' comments:

Reviewer's Responses to Questions

**Comments to the Author**

1. Is the manuscript technically sound, and do the data support the conclusions?

Reviewer #1: Yes

Reviewer #2: Yes

Reviewer #3: Yes

2. Has the statistical analysis been performed appropriately and rigorously? 

Reviewer #1: I Don't Know

Reviewer #2: I Don't Know

Reviewer #3: Yes

3. Have the authors made all data underlying the findings in their manuscript fully available?

Reviewer #1: Yes

Reviewer #2: Yes

Reviewer #3: Yes

4. Is the manuscript presented in an intelligible fashion and written in standard English?

Reviewer #1: Yes

Reviewer #2: Yes

Reviewer #3: Yes

5. Review Comments to the Author

Reviewer #1: Thank you for the opportunity to review the manuscript “Effectiveness of non-nutritive sucking on oral feeding of preterm infants: a systematic review and meta-analysis” presented in PLOS ONE

The systematic review is very interesting and will be a good contribution to share the findings once the manuscript is revised and professionally edited for English grammar and addressing the multiple sections including the recommendations for revisions and comments about the content. In the attached pdf, the feedback is detailed and specific and intended to assist with moving the manuscript to publication. Below, there is a summary of the feedback.

Best of luck

1. The title includes information about the interventions used in the systematic review, it may be added value to consider the design of included studies

2. The abstract is well defined, you may include the methods used to assess risk of bias and the synthesis of results and state the study limitations

3. Introduction:

a. Please provide an overview of preterm births in a one paragraph

b. Please provide the next section with the content specific to NNS and its benefits for premature infants

4. The aims and objectives: Please rewrite your aim and objectives based on the research questions

5. Methods:

a. The inclusion and exclusion criteria are well stated

b. Search data bases: You may refer to other databases for more studies like CINHAL

c. The outcomes measures may need to be specify as primary and secondary outcomes. Some outcomes need to be well defined along with the units of measures

6. Discussion: Please revise the discussion part, you may need to summarize it. there are too many information which may be not pertinent to the study aims.

7. Figures: Needs labelling

8. References: Please verify the citation for “Vildan 2017”, it refers to

GENERAL COMMENT

The manuscript needs to be edited to improve the flow and the content. Please refer to the feedback in the attached pdf

Reviewer #2: PONE-D-23-30243

Effectiveness of non-nutritive sucking on oral feeding of preterm infants: a systematic review and meta-analysis

PLOS ONE

Thank you for asking me to review this study.

1. Please adjust the keywords according to the mesh.

2. The most important criteria for entering and exiting the study should be mentioned in the method section of the abstract.

3. Please add the search thermals key words in the method section of the abstract if there is no word limit.

4. The time period of the search (from the beginning to the end) should be fully stated.

5. The title or research question of a systematic review should be designed based on PICO, so the comparison group, which is the same as the control group, should be added in the title.

Reviewer #3: This is a well-formed systematic review and meta-analysis focused on the fruitful subject of the effects of NNS on feeding-dependent outcomes in NICU-infants. However, there are some points which should be met before any decision for publication. These are as follows:

Abstract

1- Although the authors provided a background for the ways established for the preterm infants to achieve complete oral feeding with an implicit emphasis on Non-nutritive sucking (NNS), it seems necessary to provide a clear statement of the main questions the review addresses.

2- Also it seems necessary to provide the eligibility criteria in the method section of the abstract.

3- I propose that the PROSPERO registration be transferred inside the manuscript (may be as a footnote) out of the abstract.

4- Please provide the date of your last search for each database.

5- How did you assess risk of bias of the included studies? This needs to be clarified in the method section of the abstract.

6- Also we need specification of the methods of synthesizing the results of the systematic review.

7- Was there any risk of bias, contradiction, or ill-formed designs in the included studies which limited your conclusion? If yes, please provide a description in the conclusion of the abstract section.

Introduction

1- Page 3- line 60: I think it is better to eliminate “Currently” from the sentence, because this is not a new issue in the NICUs.

2- Page 5-line 90: In the sentence, “Although a few published meta-analyses have reached a consensus, in recent years, additional randomized controlled trials have been conducted, so we decided to carry out a more comprehensive and systematic literature review…” the authors tried to show the gap in existing knowledge of reviewed and synthesized data, however, they do not specify what agreement the previous works have reached. It is necessary to clarify this.

Methods

1- The time range of searching for sources has not been specified. We know the end not the beginning date of search.

2- Page 5-line 105: Please eliminate “were conducted”; it does make confusion in the meaning.

3- What were the eligibility criteria for the systematic review? It is described just for the meta-analysis phase.

4- Please clarify how two authors (ZSL and JHM) extracted the information like first author’s name, etc. Was it done independently too? Did they work together?

5- Page 8-line 175: the authors did not specify that how they chose 26 studies for qualitative analysis and then how they decided to do quantitative synthesis on the 21 finalized studies. On the other words, by what criteria did the authors select 21 final studies for meta-analysis? Why was 5 study excluded and 21 studies included?

6- Some of outcomes like time taken to achieve exclusive oral feeding and length of hospitalization are operationally defined. However, the others like gastrointestinal complications, time to return to birth weight, and weight at full oral feeding need a definition. We need to know the unit of their measurement (day, pound, etc.) in addition to a description of the concept. For “time to return to birth weight” specifically it is necessary to elucidate the days, as we encounter premature not term newborns.

7- The authors stated that I2≥50% indicated significant heterogeneity between studies, this is while references like Cochrane consider I2 >75% as evidence for considerable heterogeneity. Would you please cite your source?

8- Did the authors assess confidence (certainty) of the evidence for the measured outcomes?

Results

1- We do not see any trace of the qualitative analysis of the 26 chosen studies. We should have a demographic table for 26 articles first. Then the authors may go to the next step, i.e., meta-analysis of the 21 final studies. We have a missing circle here: systematic review results. Also it is necessary to explain why these 5 studies were removed from the synthesis.

2- For figures 3 to 9, I propose to mention “favors NNS-favors control” instead of “NNS-control” beneath the figures.

3- Page 10-line 212: For “weight at discharge” outcome, why didn’t the authors include references number 31, 34, and 48 in meta-analysis? These studies had calculated this secondary outcome either. Please clarify the reason of the drop out of these references from the meta-analysis.

Discussion

1- You have noted about the controversies regarding negative impact of NNS on infant health. Please explain about it. What are the probable side effects of NNS?

2- The authors attribute the lower weight of NNS group to their earlier discharge from NICU. As far as I am involved in Asadollahpour et al. (2015) study, the mean weight of the NNS group had been higher than the other groups from the baseline measurement. This may be the reason for their higher weight at discharge. This might be controlled through weight-adjustment of the groups at baseline (which was not considered at that time). Yet, as the baseline information is available in the paper, the authors may refer to it.

3- What are the probable impacts of the present review on the future research, clinical work and health policies?

6. PLOS authors have the option to publish the peer review history of their article (what does this mean?). If published, this will include your full peer review and any attached files.

Reviewer #1: No

Reviewer #2: No

Reviewer #3: No

---

## [Author Response · Author response to Decision Letter 0]

24 Feb 2024

Dear Reviewers and Editor,

Thank you very much for your reviews on our manuscript submitted to PLOS ONE and the opportunity to revise and resubmit the manuscript for further publication consideration.

We would like to thank for your time and effort to help us improve the manuscript. Each comment brought important help to improve and refine this paper. We have revised the manuscript in response to your comments and have highlighted the changes in the manuscript. Here, we respond to the reviewers' comments point by point after the comments.

Best regards,

All authors

Review Comments to the Author

Reviewer #1: Thank you for the opportunity to review the manuscript “Effectiveness of non-nutritive sucking on oral feeding of preterm infants: a systematic review and meta-analysis” presented in PLOS ONE

The systematic review is very interesting and will be a good contribution to share the findings once the manuscript is revised and professionally edited for English grammar and addressing the multiple sections including the recommendations for revisions and comments about the content. In the attached pdf, the feedback is detailed and specific and intended to assist with moving the manuscript to publication. Below, there is a summary of the feedback.

Response: We thank the reviewer’s positive comment. We would like to express our sincere gratitude to you for detailed and constructive comments. Here, we respond to the comments point by point and reply in blue after the comments.

1. The title includes information about the interventions used in the systematic review, it may be added value to consider the design of included studies

Response: Thanks to the reviewer's comment. We have amended the title “Effects of non-nutritive sucking on developmental progress and oral feeding outcomes in preterm infants: a systematic review and meta-analysis”.

2. The abstract is well defined, you may include the methods used to assess risk of bias and the synthesis of results and state the study limitations

Response: We have refined the abstract by adding methods for assessing risk of bias and outcome synthesis and explaining the limitations of this study.

3. Introduction:

a. Please provide an overview of preterm births in a one paragraph

Response: Thanks to the reviewer's comment. We have added an explanation of the circumstances of preterm birth. (Lines75-82)

b. Please provide the next section with the content specific to NNS and its benefits for premature infants

Response: We further added specifics about the benefits of NNS for preterm infants.

4.The aims and objectives: Please rewrite your aim and objectives based on the research questions

Response: We rewrite the aim and objectives of the study based on the research questions. (Lines123-128)

5. Methods:

a. The inclusion and exclusion criteria are well stated

Response: We further refined and clarified the inclusion and exclusion criteria

b. Search data bases: You may refer to other databases for more studies like CINHAL

Response: We referenced the CINHAL database and updated our search times.

c. The outcomes measures may need to be specify as primary and secondary outcomes. Some outcomes need to be well defined along with the units of measures

Response: The primary and secondary outcomes of the study we defined and added units of measurement.

6. Discussion: Please revise the discussion part, you may need to summarize it. there are too many information which may be not pertinent to the study aims.

Response: Thank you for your suggestion, we have made changes to the discussion section to remove some of the information as appropriate.

7. Figures: Needs labelling

Response: We added labelling to the figures.

8. References: Please verify the citation for “Vildan 2017”, it refers to

Response: After checking, "Vildan" is a reference to "Kaya", which we have modified.

GENERAL COMMENT The manuscript needs to be edited to improve the flow and the content. Please refer to the feedback in the attached pdf

Respond to comments in the PDF

Abstract Please add the aim of the study Response: we add the aim of study: "This review article aimed to explore the effect of NNS on developmental progress and oral feeding in preterm infants through a meta-analysis. "

Please review the total is 1339 

Response: We have re-conducted a systematic and comprehensive search of the database, which has been extended until January 2024. A total of 23 randomized controlled trials with 1461 preterm infants were included. 

Introduction

You may need to revise this sentence, it is better to start with general introduction about preterm infants and after you will specify about problems of prematurity among those feeding difficulties 

Response: We have modified the first paragraph of the preface to provide a general introduction to preterm infants, followed by a specific description of oral feeding difficulties in preterm infants. (lines75-90)

Schneider summarized in this article the importance of nutrition on the brain and cognitive development. You may need to refer to other studies to elaborate about the cardiopulmonary system and other systems 

Response: We have revised this sentence and referenced other references to make it more informative. (lines84-88)

what do you mean by motion? you may explain here the NOMAS scale 

Response: "Motion" refers to oral motor skills and is usually used to describe the coordination of the muscles of an infant while sucking, swallowing, and breathing.

We explain the NOMAS scale, the tool used to assess sucking ability in newborns.

You may need to redefine the process sine it is unclear as stated 

Response: We have redefined NNS to make its expression clearer. " NNS refers to a pacifier is placed in an infant's mouth to increase sucking action, but there is no milk or other fluid intake. "

You may elaborate this sentence 

Response: We've made changes.

It is better to discuss the type of in-consensus and support with evidence 

Response: Based on your suggestions, we have made modifications, discussed the types of opinions that are consistent with this study, and given evidence to support them.

To compare to what 

Response: NNS vs. no NNS compared

Please can you define gastrointestinal complications and related conditions

Response: Our research indicators for this study are defined in the “Materials and Methods” section. (lines161-169)

Materials and methods

You may need to look at other databases for inclusion of related studies like CINHAL, Medline and other

Response: Based on your suggestions, we searched CINHAL, Medline, and re-searched other databases.

Please add the full words for abbreviation

Response: We've added the full name of the abbreviation

the population is preterm infants less than 33 weeks of gestation, you can include in search terms the Extreme low birth weight 

Response: Our study was not limited to preterm infants <33 weeks of gestation; the inclusion criterion was preterm infants <37 weeks of gestation.

please add the definition for remaining outcomes and specify the unit of measures weight at discharge, weight at full oral feeding, time to return to birth weight and GI complications 

Response: We defined the outcomes "weight at discharge, weight at full oral feeding, time to return to birth weight, and feeding intolerance" and added units of measurement.

Results

Please review the number of studies 15 or 16

Response: We read references 15 and 16 and reevaluated the quality of the studies included in this review

please review the total 

Response: A total of 15 studies involving 942 preterm infants were included in the "time taken to achieve exclusive oral feeding”.

Subgroup analyses was performed based on gestational age of preterm infants

Please can you justify why the subgroup analyse was done based on < 33 weeks not less?

Response: The choice of subgroup analyses to be performed based on < 33 weeks gestational age rather than a younger gestational age was determined by a number of reasons. ① Infants with a gestational age of < 33 weeks are considered extremely preterm. They represent a group of particularly biologically immature infants with underdeveloped organ systems that require specialized medical care. This gestational age group is an important cut-off point for identifying very high-risk infants in clinical practice. ②For less heterogeneity. Including all preterm infants together in the analysis resulted in a study with higher heterogeneity. Therefore, by focusing on a more homogenous subgroup, researchers can reduce this heterogeneity, which can result in clearer and more precise study conclusions.

“Lighter” You may use another used for better clarity

Response: We made changes to make the expression more fluent and clearer:” The meta-analysis revealed a statistically significant difference between the two groups, with preterm infants in the NNS group exhibiting reduced weight as compared to control group.”

Discussion

You may need to refer to the article to review the result of the study about the benefit of NNS in improving sucking skills and shortening the time to transition to full breastfeeding 

Response: Thanks to the reviewer's comment. We made reference to this literature and revised the sentence.

“pure” Please can you explain what does it mean?

Response: Thanks to the reviewer's comment. " pure " means "exclusive oral feeding ".

“Tolppola et al” Add year here like Tolppola et al (2022)

Response: We have added year of Tolppola et al.

Please check the citation for the author's name 

Response: We've changed "Vaildan" to "Kaya".

nasal feeding

Response: We've changed "nasal feeding" to "tube feeding".

“There should be good communication between hospitals, communities and families to safeguard the lives and health of preterm infants.” I think this sentence is abrupt, you may need to elaborate more and add citation

Response: Thanks to the reviewer's comment. After discussion in our team group, we also felt that this sentence appeared out of place and was therefore deleted.

You may need to define the ideal behavior state

Response: NNS can help infants shift from a state of restlessness before eating to a state of quiet alertness.

You may need to review the article since sedation was not studied 

Response: We incorporated new research and revised this sentence.

You may add the strengths of the study 

Response: We added the strengths of this study. (lines403-413)

Reviewer #2:

1. Please adjust the keywords according to the mesh.

Response: Thanks to the reviewer's comment. We have adapted the keywords to the Mesh subject word.

2. The most important criteria for entering and exiting the study should be mentioned in the method section of the abstract.

Response: We mentioned the most important criteria for inclusion and exclusion in the methods section of the abstract, including:" Eligibility criteria involved randomized controlled studies in English or Chinese. Studies were excluded if they were reviews, case reports, or observational studies from which valid data could not be extracted or outcome indicators were poorly defined. "

3. Please add the search thermals key words in the method section of the abstract if there is no word limit.

Response: We added keywords for the search in the methods section of the abstract: 'non-nutritive sucking' 'Oral feeding' and 'premature.' 

4.The time period of the search (from the beginning to the end) should be fully stated.

Response: We refined the search time of the databases in the methods section of the abstract: the search period for each database was from the creation of the database until January 20, 2024.

5.The title or research question of a systematic review should be designed based on PICO, so the comparison group, which is the same as the control group, should be added in the title.

Response: Thanks to the reviewer's comment. We have made changes to the article title: "Effects of non-nutritive sucking on developmental progress and oral feeding outcomes in preterm infants: a systematic review and meta-analysis".

Reviewer #3: This is a well-formed systematic review and meta-analysis focused on the fruitful subject of the effects of NNS on feeding-dependent outcomes in NICU-infants. However, there are some points which should be met before any decision for publication. These are as follows:

Abstract

1- Although the authors provided a background for the ways established for the preterm infants to achieve complete oral feeding with an implicit emphasis on Non-nutritive sucking (NNS), it seems necessary to provide a clear statement of the main questions the review addresses.

Response：Thanks to the reviewer's comment. This review aimed to explore the effect of NNS on developmental progress and oral feeding in preterm infants through a meta-analysis.

2- Also it seems necessary to provide the eligibility criteria in the method section of the abstract.

Response：We provide some of the eligibility and exclusion criteria in the methods section of the abstract.

3- I propose that the PROSPERO registration be transferred inside the manuscript (may be as a footnote) out of the abstract.

Response：We have removed the PROSPERO registration from the abstract, which can be found under "Materials and methods" in the manuscript .(line146)

4- Please provide the date of your last search for each database.

Response：All databases last searched on January 20, 2024

5- How did you assess risk of bias of the included studies? This needs to be clarified in the method section of the abstract.

Response: "We evaluated the methodological quality of randomized controlled trials (RCTs) utilizing the assessment tool provided by the Cochrane Collaboration. " Readers can understand the methods and standards of studying bias risk assessment when reading the abstract.

6- Also we need specification of the methods of synthesizing the results of the systematic review.

Response: We describe this in the methods section of the abstract. " The meta-analysis will utilize Review Manager 5.3 software, employing either random-effects or fixed-effects models based on observed heterogeneity. We calculated the mean difference (MD) and 95% confidence interval (CI) for continuous data, and estimated pooled odds ratios (ORs) for dichotomous data. "

7- Was there any risk of bias, contradiction, or ill-formed designs in the included studies which limited your conclusion? If yes, please provide a description in the conclusion of the abstract section.

Response: Yes, there are some limitations in this study. We describe it in the conclusion of the abstract. " The study, hampered by a small sample size, did not account for potential confounders. High heterogeneity and notable bias among studies were observed. "

Introduction

1- Page 3- line 60: I think it is better to eliminate “Currently” from the sentence, because this is not a new issue in the NICUs.

Response: We have eliminated “Currently”.

2- Page 5-line 90: In the sentence, “Although a few published meta-analyses have reached a consensus, in recent years, additional randomized controlled trials have been conducted, so we decided to carry out a more comprehensive and systematic literature review…” the authors tried to show the gap in existing knowledge of reviewed and synthesized data, however, they do not specify what agreement the previous works have reached. It is necessary to clarify this.

Response: We interpret the consensus reached in previously published review studies that non-nutritive sucking improves feeding skills and reduces hospitalization time in preterm infants.

Methods

1- The time range of searching for sources has not been specified. We know the end not the beginning date of search.

Response: The search time frame for each database ranged from its inception date until January 20, 2024.

2- Page 5-line 105: Please eliminate “were conducted”; it does make confusion in the meaning.

Response: We have eliminated “were conducted”.

3- What were the eligibility criteria for the systematic review? It is described just for the meta-analysis phase.

Response: The eligibility criteria for the systematic review were consistent with those for the meta-analysis, and we have refined them.

4- Please clarify how two authors (ZSL and JHM) extracted the information like first author’s name, etc. Was it done independently too? Did they work together?

Response: The two authors (ZSL and JHM) extracted general information for inclusion in the study independently, and they did not work together and did not interfere with each other. 

5- Page 8-line 175: the authors did not specify that how they chose 26 studies for qualitative analysis and then how they decided to do quantitative synthesis on the 21 finalized studies. On the other words, by what criteria did the authors select 21 final studies for meta-analysis? Why was 5 study excluded and 21 studies included?

Response: To ensure the comprehensiveness and up-to-date nature of our research, we have updated the literature search cutoff date to incorporate a newly published study. This article offers a detailed account of the selection process for the final 22 studies included. In the ultimate phase of this process, five studies were excluded due to insufficient quality.

6- Some of outcomes like time taken to achieve exclusive oral feeding and length of hospitalization are operationally defined. However, the others like gastrointestinal complications, time to return to birth weight, and weight at full oral feeding need a definition. We need to know the unit of their measurement (day, pound, etc.) in addition to a description of the concept. For “time to return to birth weight” specifically it is necessary to elucidate the days, as we encounter premature not term newborns.

Response: We defined time to return to birth weight and weight at full oral feeding, as well as feeding intolerance, and added their units of measurement.

7- The authors stated that I2 ≥50% indicated significant heterogeneity between studies, this is while references like Cochrane consider I2 >75% as evidence for considerable heterogeneity. Would you please cite your source?

Response: We refer to the research of Foreman et al and think that I2≥50% indicated significant heterogeneity between studies.

(Foreman J, Salim AT, Praveen A, et al. Association between digital smart device use and myopia: a systematic review and meta-analysis. Lancet Digit Health. 2021;3(12):e806-e818. doi:10.1016/S2589-7500(21)00135-7)

8- Did the authors assess confidence (certainty) of the evidence for the measured outcomes?

Response: Yes, the authors assessed confidence (certainty) of the evidence for the measured outcomes. We assessed the methodological quality of RCTs using the Cochrane risk of bias tool, which included seven evaluation items: sequence generation, allocation concealment, blinding of participants, outcome evaluator, incomplete outcome data, and selective outcome reporting. The studies were evaluated for unclear, low or high risk of bias.

Results

1- We do not see any trace of the qualitative analysis of the 26 chosen studies. We should have a demographic table for 26 articles first. Then the authors may go to the next step, i.e., meta-analysis of the 21 final studies. We have a missing circle here: systematic review results. Also it is necessary to explain why these 5 studies were removed from the synthesis.

Response: We qualitatively analyzed the 28 studies in Table 1 and provided detailed explanations for these 5 studies that were removed from the review. (See Table 1 for details.) Four of the five studies that were deleted were because the control group also received NNS or pacifier interventions, and one study was deleted because the study metrics could not be converted for data.

2- For figures 3 to 9, I propose to mention “favors NNS-favors control” instead of “NNS-control” beneath the figures.

Response: We have revised figures 3 to 9 to “favors NNS-favors control” based on your suggestions.

3- Page 10-line 212: For “weight at discharge” outcome, why didn’t the authors include references number 31, 34, and 48 in meta-analysis? These studies had calculated this secondary outcome either. Please clarify the reason of the drop out of these references from the meta-analysis.

Response: Reference 31, which measured from "admission to weight of 2 kg," did not match our indicator of "weight at discharge" and was deleted.

References 34 and 48, although they had "weight at discharge" results, were included in the analysis and found to be the main source of heterogeneity, and were deleted. We have explained this in the manuscript in the "weight at discharge" results. (lines262-268)

Discussion

1- You have noted about the controversies regarding negative impact of NNS on infant health. Please explain about it. What are the probable side effects of NNS?

Response: NNS provided by pacifiers tend to cause nipple confusion and may also cause different physiological, pharyngeal, and esophageal motility events, potentially affecting gastroesophageal reflux.

2- The authors attribute the lower weight of NNS group to their earlier discharge from NICU. As far as I am involved in Asadollahpour et al. (2015) study, the mean weight of the NNS group had been higher than the other groups from the baseline measurement. This may be the reason for their higher weight at discharge. This might be controlled through weight-adjustment of the groups at baseline (which was not considered at that time). Yet, as the baseline information is available in the paper, the authors may refer to it.

Response: Thanks to the reviewer's suggestion, we reviewed the baseline data of the included studies and found that the reason for the difference in "weight at discharge" between the studies may be related to weight at baseline. We have made changes in the "Discussion" section of the manuscript.

3- What are the probable impacts of the present review on the future research, clinical work and health policies?

Response: This review conducted an extensive literature search to refine the evidence base supporting the use of NNS to improve oral feeding outcomes in preterm infants and to promote evidence-based feeding practices. The findings of this study underscore the benefits of NNS for oral feeding in preterm infants, which can inform the development of structured oral feeding programs in NICUs. Such programs have the potential to provide preterm infants with positive early feeding experiences, thereby enhancing their overall care and development. Moving forward, there is a need for large-scale multicenter randomized controlled trials to further investigate the efficacy of NNS in improving oral feeding in preterm infants, and to assist in the development of individualized feeding patterns and evidence-based clinical guidelines for oral feeding in preterm infants.

---

## [Decision Letter · Decision Letter 1]

22 Mar 2024

PONE-D-23-30243R1Effects of non-nutritive sucking on developmental progress and oral feeding outcomes in preterm infants: a systematic review and meta-analysisPLOS ONE

Dear Dr. cui,

Thank you for submitting your manuscript to PLOS ONE. After careful consideration, we feel that it has merit but does not fully meet PLOS ONE’s publication criteria as it currently stands. Therefore, we invite you to submit a revised version of the manuscript that addresses the points raised during the review process.

We look forward to receiving your revised manuscript.

Kind regards,

Mona Nabulsi, MD, MS

Academic Editor

PLOS ONE

Journal Requirements:

Reviewer's Responses to Questions

**Comments to the Author**

1. If the authors have adequately addressed your comments raised in a previous round of review and you feel that this manuscript is now acceptable for publication, you may indicate that here to bypass the “Comments to the Author” section, enter your conflict of interest statement in the “Confidential to Editor” section, and submit your "Accept" recommendation.

Reviewer #1: All comments have been addressed

Reviewer #2: All comments have been addressed

Reviewer #3: (No Response)

2. Is the manuscript technically sound, and do the data support the conclusions?

Reviewer #1: Yes

Reviewer #2: Yes

Reviewer #3: Yes

3. Has the statistical analysis been performed appropriately and rigorously? 

Reviewer #1: I Don't Know

Reviewer #2: I Don't Know

Reviewer #3: Yes

4. Have the authors made all data underlying the findings in their manuscript fully available?

Reviewer #1: Yes

Reviewer #2: Yes

Reviewer #3: Yes

5. Is the manuscript presented in an intelligible fashion and written in standard English?

Reviewer #1: Yes

Reviewer #2: Yes

Reviewer #3: Yes

6. Review Comments to the Author

Reviewer #1: Review 16/3/2024

Thank you for your extensive work and addressing the suggestions and comments. The manuscript demonstrated thorough and diligent efforts

Kindly find below minor suggestions for your kind follow up best of luck

1- Abstract :

a. Title : Based on your research question you may need to modify the title as: “Effects Of Implementing Non-Nutritive Sucking On Oral Feeding Progression And Outcomes In Preterm Infants: A Systematic Review And Meta-Analysis”

b. Aim: Line 34-35 You may need to review the aim “to explore the effect of NNS on oral feeding progression through a meta-analysis”.

c. Conclusion: Line 64 you may need to modify the sentence to “reduces the time to reach full oral feeding”

2- Introduction: Line 88-90 the cited reference discusses the effect of oxygen therapy on altering the oral sensory, motor, and the development of coordinated NNS in preterm infants. I may suggest reviewing another reference to support your point.

3- Materials and methods

a. Systematic search and strategy

Line 143 -1 44 “Detailed search strategies for each database are shown in S2 Table” you may need to modify to S1 table as stated in the section of supporting information.

b. Inclusion and exclusion criteria: Line 157-165: You may need to put references for the definition of outcome measures and add the full word of the unit “g” as grams

4- Results: Line 215 - 227: Please review the values in the related section and match with ones stated in Fig1. PRISMA flow diagram

5- Discussion:

a. Line 325, please review the citation, it is more pertinent for the reference (57)

“Barlow SM, Finan DS, Rowland SG. Mechanically evoked perioral reflexes in infants. Brain research. 651 1992;599(1):158-60. Epub 1992/12/18. doi: 10.1016/0006-8993(92)90865-7. PubMed PMID: 1493544.”

b. Line 347, The author Chorna et al studied the outcome for the development of sucking ability and decrease the length of hospitalization. I may suggest looking at this reference instead which elaborates the complex factors for discharge “Committee on Fetus and Newborn. (2008). Hospital discharge of the high-risk neonate. Pediatrics, 122(5), 1119-1126.”

Reviewer #2: PONE-D-23-30243

Effectiveness of non-nutritive sucking on oral feeding of preterm infants: a systematic review and meta-analysis

PLOS ONE

Thank you for asking me to review this study. The authors' answers were acceptable to me and convincing.

Thank you

Reviewer #3: (No Response)

7. PLOS authors have the option to publish the peer review history of their article (what does this mean?). If published, this will include your full peer review and any attached files.

Reviewer #1: **Yes: **Saadieh Masri

Reviewer #2: No

Reviewer #3: No

---

## [Author Response · Author response to Decision Letter 1]

25 Mar 2024

Reviewer #1:  Review 16/3/2024

Thank you for your extensive work and addressing the suggestions and comments. The manuscript demonstrated thorough and diligent efforts

Kindly find below minor suggestions for your kind follow up best of luck

Response: Once again, we express our appreciation for your constructive comments. Here, we have responded to the comments individually in blue.

1- Abstract:

a. Title: Based on your research question you may need to modify the title as: “Effects Of Implementing Non-Nutritive Sucking On Oral Feeding Progression And Outcomes In Preterm Infants: A Systematic Review And Meta-Analysis”

b. Aim: Line 34-35 You may need to review the aim “to explore the effect of NNS on oral feeding progression through a meta-analysis”.

c. Conclusion: Line 64 you may need to modify the sentence to “reduces the time to reach full oral feeding”

Response: Thanks to the reviewer's comment. 

a.As per your suggestion, I have revised the title to " Effects Of Implementing Non-Nutritive Sucking On Oral Feeding Progression And Outcomes In Preterm Infants: A Systematic Review And Meta-Analysis." 

b.The purpose of the study in the abstract section has been revised to: "This review aimed to explore the effect of NNS on oral feeding progression through a meta-analysis. " 

c.The first sentence of the conclusion has been modified as per your suggestion to read " NNS improves oral feeding outcomes in preterm infants and reduces the time to reach full oral feeding and hospitalization length."

2- Introduction: Line 88-90 the cited reference discusses the effect of oxygen therapy on altering the oral sensory, motor, and the development of coordinated NNS in preterm infants. I may suggest reviewing another reference to support your point.

Response: We have added other references to support our argument.

 (9. Samara M, Johnson S, Lamberts K, Marlow N, Wolke D. Eating problems at age 6 years in a whole population sample of extremely preterm children. Developmental medicine and child neurology. 2010;52(2):e16-22. Epub 2009/10/17. doi: 10.1111/j.1469-8749.2009.03512.x. PubMed PMID: 19832883.)

3- Materials and methods

a. Systematic search and strategy

Line 143 -1 44 “Detailed search strategies for each database are shown in S2 Table” you may need to modify to S1 table as stated in the section of supporting information.

b. Inclusion and exclusion criteria: Line 157-165: You may need to put references for the definition of outcome measures and add the full word of the unit “g” as gram

Response: Thanks to the reviewer's comment. 

a. See S1 Table for detailed search strategies for each database, which we have modified.

b. We have added references to all the definitions for the outcome measures and modified the unit "g" to "gram". (lines 159-168)

4- Results: Line 215 - 227: Please review the values in the related section and match with ones stated in Fig1. PRISMA flow diagram

Response: Lines 216-228, we checked this section well compared to the values in Figure 1 (PRISMA flow diagram) and verified that there were no errors.

5- Discussion:

a. Line 325, please review the citation, it is more pertinent for the reference (57)

“Barlow SM, Finan DS, Rowland SG. Mechanically evoked perioral reflexes in infants. Brain research. 651 1992;599(1):158-60. Epub 1992/12/18. doi: 10.1016/0006-8993(92)90865-7. PubMed PMID: 1493544.”

Response: We amend the citation in line 325 to read "59. Barlow SM, Finan DS, Rowland SG. Mechanically evoked perioral reflexes in infants. Brain research. 1992;599(1):158-60. Epub 1992/12/18. doi: 10.1016/0006-8993(92)90865-7. PubMed PMID: 1493544."

b. Line 347, The author Chorna et al studied the outcome for the development of sucking ability and decrease the length of hospitalization. I may suggest looking at this reference instead which elaborates the complex factors for discharge “Committee on Fetus and Newborn. (2008). Hospital discharge of the high-risk neonate. Pediatrics, 122(5), 1119-1126.”

Response: As you suggested, we have carefully read and quoted from the article (62. Hospital discharge of the high-risk neonate. Pediatrics. 2008;122(5):1119-26. Epub 2008/11/04. doi: 10.1542/peds.2008-2174. PubMed PMID: 18977994.) and have refined this section of the manuscript. (lines 346-352)

Reviewer #2: Thank you for asking me to review this study. The authors' answers were acceptable to me and convincing.

Response: We are pleased to learn that you found our response acceptable and convincing. Thank you again for your interest in and support of our research, and good luck.

We would like to thank the reviewer again for your time and efforts to provide the comments for us to improve the manuscript.

---

## [Editor Report · Decision Letter 2]

1 Apr 2024

Effects Of Implementing Non-Nutritive Sucking On Oral Feeding Progression And Outcomes In Preterm Infants: A Systematic Review And Meta-Analysis

PONE-D-23-30243R2

Dear Dr. cui,

We’re pleased to inform you that your manuscript has been judged scientifically suitable for publication and will be formally accepted for publication once it meets all outstanding technical requirements.

Kind regards,

Mona Nabulsi, MD, MS

Academic Editor

PLOS ONE

---

## [Editor Report · Acceptance letter]

4 Apr 2024

PONE-D-23-30243R2 

PLOS ONE

Dear Dr. Cui, 

I'm pleased to inform you that your manuscript has been deemed suitable for publication in PLOS ONE. Congratulations! Your manuscript is now being handed over to our production team.

Kind regards, 

on behalf of

Dr. Mona Nabulsi 

Academic Editor

PLOS ONE